# Effects of Different Guests on Pyrolysis Mechanism of α-CL−20/Guest at High Temperatures by Reactive Molecular Dynamics Simulations at High Temperatures

**DOI:** 10.3390/ijms24031840

**Published:** 2023-01-17

**Authors:** Mingming Zhou, Jing Luo, Dong Xiang

**Affiliations:** College of Chemistry and Environmental Engineering, Yangtze University, Jingzhou 434023, China

**Keywords:** host-guest inclusion strategy, pyrolysis decomposition, ReaxFF-MD, reaction rate

## Abstract

The host–guest inclusion strategy has the potential to surpass the limitations of energy density and suboptimal performances of single explosives. The guest molecules can not only enhance the detonation performance of host explosives but also can enhance their stability. Therefore, a deep analysis of the role of guest influence on the pyrolysis decomposition of the host–guest explosive is necessary. The whole decomposition reaction stage of CL-20/H_2_O, CL-20/CO_2_, CL-20/N_2_O, CL-20/NH_2_OH was calculated by ReaxFF-MD. The incorporation of CO_2_, N_2_O and NH_2_OH significantly increase the energy levels of CL-20. However, different guests have little influence on the initial decomposition paths of CL-20. The *E*_a1_ and *E*_a2_ values of CL-20/CO_2_, CL-20/N_2_O, CL-20/NH_2_OH systems are higher than the CL-20/H_2_O system. Clearly, incorporation of CO_2_, N_2_O, NH_2_OH can inhibit the initial decomposition and intermediate decomposition stage of CL-20/H_2_O. Guest molecules become heavily involved in the reaction and influence on the reaction rates. *k*_1_ of CL-20/N_2_O and CL-20/NH_2_OH systems are significantly larger than that of CL-20/H_2_O at high temperatures. *k*_1_ of CL-20/CO_2_ system is very complex, which can be affected deeply by temperatures. *k*_2_ of the CL-20/CO_2_, CL-20/N_2_O systems is significantly smaller than that of CL-20/H_2_O at high temperatures. *k*_2_ of CL-20/NH_2_OH system shows little difference at high temperatures. For the CL-20/CO_2_ system, the *k*_3_ value of CO_2_ is slightly higher than that for CL-20/H_2_O, CL-20/N_2_O, CL-20/NH_2_OH systems, while the *k*_3_ values of N_2_ and H_2_O are slightly smaller than that for the CL-20/H_2_O, CL-20/N_2_O, CL-20/NH_2_OH systems. For the CL-20/N_2_O system, the *k*_3_ value of CO_2_ is slightly smaller than that for CL-20/H_2_O, CL-20/CO_2_, CL-20/NH_2_OH systems. For the CL-20/NH_2_OH system, the *k*_3_ value of H_2_O is slightly larger than that for CL-20/H_2_O, CL-20/CO_2_, CL-20/N_2_O systems. These mechanisms revealed that CO_2_, N_2_O and NH_2_OH molecules inhibit the early stages of the initial decomposition of CL-20 and play an important role for the decomposition subsequently.

## 1. Introduction

Successful balance between high energy and safety of energetic materials is challenging due to the time-consuming and difficult nature of synthesizing new energetic materials. Host–guest energetic materials, as shown in Figure 1, by embedding hydrogen- [1,2,3] or nitrogen-containing [4,5] oxidizing small molecules into the crystal lattice voids may achieve the highest possible energy density and the maximum possible chemical stability [6].

2,4,6,8,10,12-hexanitro-2,4,6,8,10,12-hexaazaisowurtzitane (CL-20), as one of the energetic materials with greatest detonation performances and highest density from synthesized compounds, has been widely studied recently [7]. CL-20 has five polymorphs (namely β, γ, ε, ζ and δ) and an α-form hydrate [8], in which the ε-CL-20 has the highest density (2.044 g·cm^−3^) [8], β-CL-20 and γ-CL-20 are the dominant configurations of CL-20 molecules found in co-crystals [9]. Thus, many approaches have been used to tune its performance, such as co-crystals [10,11,12,13], nanosized particles [14] and so on. The nanoscaling cocrystals demonstrate enhanced stability and high solubility of nano-particles [15]. However, the high production costs and easily phase transition limit the widespread use of CL-20 [16,17]. Fortunately, removal of H_2_O from hydrated α-CL-20 can retain the stacking model [18]. Then, small molecules are filled in the cavity structure region that removes H_2_O molecules to form α-CL-20 host–guest explosives [3,4,5], which also can be regarded as the solvates of CL-20 [19,20,21]. Bennion J.C. [3] synthesized two polymorphic hydrogen peroxide (HP) solvates of α-CL-20. Two α-CL-20/H_2_O_2_ host–guest compounds are scarcely changing the lattice volume of the α-CL-20/H_2_O and improving the energy. Thereafter, researches are focused on adopting a host–guest inclusion strategy to embed a suitable guest within the void by removing the water of α-CL-20/H_2_O. A series of α-CL-20-guest energetic materials such as CL-20/CO_2_ [4], CL-20/N_2_O [4], CL-20/NH_2_OH [5] have been constructed in this manner.

The simulation of α-CL-20/guest mainly focused on the mechanism of host–guest molecular interaction and the pyrolysis mechanism at high temperatures. The intermolecular interaction is the central scientific issue of energetic cocrystals. Guo [6] had reviewed some typical energetic inclusion compounds and their structures, intermolecular interactions, stabilities, and energy properties. It provides a method to predict appropriate size of guest incorporate into the cavities of the α-CL-20 crystal. The systematic studies [22,23,24] on the comparison of interaction between the host–guest energetic complexes are devoted to summarizing the influence of guest on the performance of α-CL-20. The results would prove fundamental to summarize the properties of guest for α-CL-20/guest with structural stability. Meanwhile, in order to deeply analyze the role of hydrogen–guest small molecules in the host–guest system, the initial decomposition reactions of ICM-102/HNO_3_ [25], ICM-102/H_2_O_2_ [26] with pure ICM-102 and CL-20/H_2_O_2_ [27] at several high temperatures were systematically studied by molecular dynamics simulations. It was found that the addition of small guest molecules significantly increased the energy levels of ICM-102 and CL-20 but had little effect on the thermal stability of the host-guest system. The initial reaction path of ICM-102 molecule was not changed by HNO_3_ and H_2_O_2_, but HNO_3_ and H_2_O_2_ promoted the decomposition of ICM-102 molecule in the subsequent decomposition process. With the increase of temperature, the influence of H_2_O_2_ on the pyrolysis reaction of CL-20 weakens [28]. 

All the simulation researches provide information to understand the influence of guest for the host explosives. However, the role of different nitrogen-guest small molecules in the host–guest system has not been studied systematically at different high temperatures. At the same high temperature, when do the different guest molecules participate in the decomposition reaction of host–guest explosive and how do they affect the decomposition process mechanism of host–guest explosive? What is the influence of the same guest on the pyrolysis of the host–guest explosive at different high temperatures? Therefore, detailed studies of the mechanism of the α-CL-20/nitrogen-guest detonation reaction at different high temperatures are necessary.

ReaxFF-MD [25,27,28] can conduct in-depth and detailed research on the pyrolysis mechanism of host–guest explosive at the microscopic scale and find how guest molecules participate in the pyrolysis reaction of the guest is an important factor affecting the energy release and detonation performance of the host explosives. In this study, we investigated the initial reaction of CL-20/CO_2_, CL-20/N_2_O, CL-20/NH_2_OH and compared with the pure CL-20/H_2_O at various temperatures (2500, 2750, 3000, 3250, and 3500 K) by ReaxFF-lg reactive MD simulations (MD/ReaxFF-lg). The initial reaction paths, the change of generated/destroyed chemical bond numbers, the main product compositions, kinetic parameters in the different stages were analyzed. The mechanism for the improvement of the explosive energy and stability by incorporation of CO_2_, N_2_O and NH_2_OH is also discussed. 

## 2. Results and Discussion

### 2.1. Potential Energy (PE) and Total Energy for CL-20/H_2_O, CL-20/CO_2_, CL-20/N_2_O, CL-20/NH_2_OH Systems

The evolution of potential energy (PE) of CL-20/H_2_O, CL-20/CO_2_, CL-20/N_2_O, CL-20/NH_2_OH systems with time at (a) 2500 K, (b) 2750 K, (c) 3000 K, (d) 3500 K is shown in Figure 2. All the PE of CL-20/guest are much larger than that of CL-20/H_2_O. It demonstrates that the addition of small guest molecules significantly increases the energy levels of CL-20 just shown in Figure 3. All the systems exhibit an initial rise in the PE curves at different temperatures which correspond to the endothermic reaction stage. When PE is maximized, the value thereafter decreases, signifying that the reaction becomes exothermic. The maximum value of PE increases in the order incorporation of H_2_O < NH_2_OH < CO_2_ < N_2_O at different temperatures, and the heat release is also increased. That is, the incorporation of guests may have remarkable influence on heat release during the reaction. At a relatively low temperature of 2500 K, the PE curve is smooth. However, when the temperature increases to 2750 K, the PE curve is a little bit steeper. As the temperature increases to, much higher, 3000 K and 3500 K, the PE curve changes little. That is, no obvious heat release occurs during the reaction for much higher temperatures. With the increase of temperature, the PE is observed to be close to equilibrium for much shorter time. Therefore, the higher the temperature, the earlier the complete reaction.

The evolution of potential energy (PE) of (a) CL-20/H_2_O, (b) CL-20/CO_2_, (c) CL-20/N_2_O, (d) CL-20/NH_2_OH system with time at different temperatures is shown in Figure 4. The trend of PE curves for CL-20/H_2_O and CL-20/N_2_O is very similar. The trend of PE curves for CL-20/CO_2_ and CL-20/NH_2_OH is very similar. That is, the incorporation of N_2_O may have the same influence on heat release with H_2_O. However, the incorporation of CO_2_ may have the same influence on heat release with NH_2_OH.

### 2.2. Initial Decomposition Stage

#### 2.2.1. Initial Reaction Path of CL-20/Nitrogen-Guest

Table 1 shows the initial reaction paths of host–guest molecules and their occurrence frequency for CL-20/H_2_O, CL-20/CO_2_, CL-20/N_2_O, CL-20/NH_2_OH at four high temperatures. There are two main initial decomposition reactions of host CL-20 molecules C_6_H_6_O_12_N_12_ → C_6_H_6_O_10_N_11_ + NO_2_ and C_6_H_6_O_12_N_12_ → C_6_H_5_O_12_N_12_ + H. The frequency of C_6_H_6_O_12_N_12_ → C_6_H_6_O_10_N_11_ + NO_2_ is much more than that of C_6_H_6_O_12_N_12_ → C_6_H_5_O_12_N_12_ + H. As the increase with the temperatures, the frequency of both main initial decomposition reaction improves for CL-20/H_2_O, CL-20/CO_2_, CL-20/N_2_O, CL-20/NH_2_OH. A small part of H_2_O, N_2_O, NH_2_OH is broken to smaller pieces except for CO_2_ with no decomposition. At the same high temperature, the frequency of both main initial decomposition reactions is not significantly different for CL-20/H_2_O, CL-20/CO_2_, CL-20/N_2_O, CL-20/NH_2_OH. It demonstrates that different guests have little influence on the initial decomposition paths.

#### 2.2.2. Effect of Nitrogen-Guest on the k_1_

There are three stages for the evolution of the thermal decomposition of CL-20/nitrogen-guest. Firstly, the initial decomposition stage is characterized by rate constant *k*_1_ and activation energy *E*_a1._ Then, the intermediate decomposition stage is characterized by rate constant *k*_2_ and activation energy *E*_a2._ Finally, the final product evolution stage is characterized by rate constant *k*_3_ and activation energy *E*_a3_.

During the initial decomposition stage, the reaction rate was calculated by the change of the number of CL-20 molecules. The decay of the number of CL-20 molecules with time follows first-order decay exponential function [29]: *N*(*t*) = *N*_0_ × exp[−*k*_1_(*t* − *t*_0_)], where *N*_0_ is the initial number of CL-20 molecules, *t*_0_ is the time when CL-20 started to decompose, and *k*_1_ is the initial decomposition stage rate constant (Table 2). 

The logarithm of *k*_1_ plotted against the inverse temperature (1/*T*) at 2500, 2750, 3000, 3250, and 3500 K is shown in Figure 5. The *E*_a1_ values of the CL-20/H_2_O, CL-20/CO_2_, CL-20/N_2_O, CL-20/NH_2_OH systems are 64.90, 87.12, 81.93 and 90.12 kJ∙mol^−1^, respectively. Clearly, incorporation of CO_2_, N_2_O, NH_2_OH impede the initial decomposition. This indicates that nitrogen-guest can inhibit the trigger decomposition of CL-20/H_2_O. 

In addition, *k*_1_ of CL-20/N_2_O system is significantly larger than that of CL-20/H_2_O at high temperatures. This indicates that N_2_O significantly accelerates the reaction rate in the initial decomposition stage at high temperatures. *k*_1_ of CL-20/NH_2_OH system is significantly larger than that of CL-20/H_2_O at relatively higher temperatures (3500 K, 3250 K, 3000 K, 2750 K). As the temperature decreased to 2500 K, the difference between *k*_1_ of the CL-20/NH_2_OH and CL-20/N_2_O systems almost disappears. This indicates that NH_2_OH significantly accelerates the reaction rate in the initial decomposition stage at relatively higher temperature. *k*_1_ of CL-20/CO_2_ system is much complex. At higher temperatures, *k*_1_ of CL-20/CO_2_ is much larger than that of CL-20/H_2_O. However, *k*_1_ of CL-20/CO_2_ is much smaller than that of CL-20/H_2_O at relatively lower temperatures. This indicates the temperature has significant influence on the initial decomposition rate for CL-20/CO_2_. 

### 2.3. Intermediate Decomposition Stage

#### 2.3.1. Effect of Nitrogen-Guest on the Main Intermediate Products

Figure 6 shows the evolution curves of the main intermediate products and host-guest molecules at different temperatures. For CL-20/H_2_O at 2500 K, the number curve of host CL-20 fluctuates slightly, but the overall level remains horizontal before 0.5 ps. The NO_2_ fragments appears immediately at about 0ps. However, the number curve of guest H_2_O fluctuates slightly, but the overall level remains horizontal before 0.9 ps. This demonstrates that the initial decomposition of CL-20/H_2_O may have broken the C–NO_2_ bonds of host CL-20 to form NO_2_. During 0.5 ps~1 ps, the number of host CL-20 decreases sharply and disappears at 1ps, while the number of NO_2_ fragments increases rapidly. During 0.9 ps~1 ps, the number of guest H_2_O decreases, while the number of guest H_2_O reaches the minimum value. It demonstrates that guest H_2_O begins to participate the decomposition reaction deeply. The results of the trend are consistent with those of PE before 1ps for endothermic decomposition stage. During 0.5 ps~1 ps, the number of guest NO_2_ increases sharply, while the number of guest NO_2_ reaches the maximum value. Due to the participation of H_2_O, the pyrolysis products begin to diversify. The NO_3_ and NO fragments begin to appear. All the curves for NO_3_ and NO fragments are similarity at the high temperatures. However, the amount of NO_3_ and NO fragments would improve as the increase of temperatures. As the temperature increased, the variation curves of the main intermediate produces and host–guest molecules for CL-20/H_2_O remains the same approximately. However, the reaction rates (*k*_2_) are significantly different. The influence of high temperature on *k*_2_ will be analyzed in the following section.

For CL-20/CO_2_, CL-20/N_2_O, CL-20/NH_2_OH at high temperatures, the evolution tendency of the main intermediate products and host molecules is similar with that for CL-20/H_2_O at 2500 K. The variation curves of the guest are quite different. 

Figure 7, Figure 8 and Figure 9 show the evolution curves of the host and guest molecules for CL-20/H_2_O, CL-20/CO_2_, CL-20/N_2_O, CL-20/NH_2_OH at 2500 K, 3000 K, 3500 K. All the host CL-20 variation tendency are similarity. The influence of different guest on *k*_2_ is not significant at 2500 K. With higher temperature, the *k*_2_ significantly larger. The guest H_2_O and CO_2_, they are the main decomposition products. The variation tendency can divide into four stages: firstly, the tendency of guest is a level for little changeable. Secondly, it decreases for guest decomposition quickly. Then, it increases quickly as the main products. Finally, it reaches horizontal for the completely decomposition. There are two differences for the variation tendency: the first is that H_2_O and CO_2_ start to decrease at different times. The longer time to stay at the first stage for CO_2_ shows that CO_2_ may be more stability than H_2_O, and the second is that the minimum values of H_2_O and CO_2_ are different. It a greater intensity in pyrolysis reaction for H_2_O than that of CO_2_. As the increase of temperatures, the shorter time to stay the first stage and the smaller minimum for CO_2_ and H_2_O. It displays that the more intense in pyrolysis reaction at higher temperatures. For N_2_O, NH_2_OH just as the role for guest, the variation tendency slowly decreases and then sharply decreases until disappears. However, the reaction rates (*k*_2_) for CO_2_, N_2_O, NH_2_OH are significantly different at different temperatures. 

#### 2.3.2. Effect of Nitrogen-Guest on the k_2_

After the PE reached the maximum value, the intermediate exothermic decomposition indicates the chemical reaction stage. The intermediate decomposition stage rate constant *k*_2_ can be obtained by fitting the PE curves with a first order decay exponential function [30]: *U*(*t*) = *U*_∞_ + Δ*U_exo_*_•_exp[−*k*_2_(*t* − *t*_max_)], where *U*(*t*) is the potential energy value at time t, *U*_∞_ is the asymptotic value of PE, Δ*U_exo_* is the reaction heat, and its size is the difference between the maximum potential energy *U*_max_ and *U*_∞._

The chemical reaction rate constants obtained by fitting equation at different temperatures are shown in Table 3. The value of Δ*U_exo_* has little change with a gradually increase of *U*_∞_ and *k*_2_ as temperature increases. This indicates that temperature has a limited effect on the exothermic reaction [31].

The logarithm of *k*_2_ plotted against the inverse temperature (1/*T*) at 2500, 2750, 3000, 3250, and 3500 K is shown in Figure 10. The *E*_a2_ values of the CL-20/H_2_O, CL-20/CO_2_, CL-20/N_2_O, CL-20/NH_2_OH systems are 80.76, 87.58, 92.73 and 84.88 kJ∙mol^−1^, respectively. Clearly, incorporation of CO_2_, N_2_O, NH_2_OH can inhibit the intermediate decomposition of CL-20/H_2_O. 

The pre-exponential factor derived from the pyrolysis simulations of the CL-20/H_2_O, CL-20/CO_2_, CL-20/N_2_O, CL-20/NH_2_OH systems are 29.65, 29.68, 30.03, 29.80. Assuming unimolecular decomposition, transition state theory leads to A = (k_B_T/h)exp(ΔS/R) where ΔS_(CL-20/H2O)_ = −17.47 J·mol^−1^·K^−1^, ΔS_(CL-20/CO2)_ = −17.23 J·mol^−1^·K^−1^, ΔS_(CL-20/N2O)_= −14.32 J·mol^−1^·K^−1^, ΔS_(CL-20/NH2OH)_ = −16.23 J·mol^−1^·K^−1^. This negative activation of entropy is consistent with the TST for multimolecular reactions, suggesting that the reaction involves a multimolecular transition state [32]. The decrease of entropy at the transition state because of the embedding of nitrogen-containing guests.

In addition, *k*_2_ of CL-20/CO_2_ system is significantly smaller than that of CL-20/H_2_O at high temperatures. This indicates that CO_2_ significantly inhibits the reaction in the intermediate decomposition stage at high temperatures. *k*_2_ of CL-20/N_2_O system is significantly smaller than that of CL-20/H_2_O at relatively lower temperatures (2500 K and 2750 K). As the temperature increased to 3500 K, the difference between *k*_2_ of the CL-20/H_2_O and CL-20/N_2_O systems almost disappears. This indicates that N_2_O significantly restrains the reaction in the intermediate decomposition stage at relatively low temperature. With increasing temperature, N_2_O has increasingly less effect on the reaction rate. However, *k*_1_ of CL-20/N_2_O indicates that N_2_O significantly accelerates the reaction in the initial decomposition stage at high temperatures. The conclusion is contrary to that for ICM-102/HNO_3_ [27]. This maybe caused by the hydrogen content for nitrogen-guest. *k*_2_ of CL-20/H_2_O and CL-20/NH_2_OH systems are little difference at high temperatures, NH_2_OH has little effect on the reaction rate at high temperatures. The opposite effect of CL-20/H_2_O_2_ [33] maybe due to the difference of hydrogen content in the guest. The influence of CO_2_ and N_2_O on the decomposition reaction of host explosive may be the little interaction between CO_2_, N_2_O and CL-20 [27]. However, the influence of H_2_O_2_ on the decomposition reaction of host explosive may be the significant interaction between H_2_O_2_ and CL-20 [27].

### 2.4. Final Product Evolution Stage

#### 2.4.1. Effect of Nitrogen-Guest on the Final Products

To clarify the effect of CO_2_, N_2_O and NH_2_OH molecules on the main products, the population of CO_2_, N_2_, H_2_O after the complete decomposition reaction for CL-20/H_2_O, CL-20/CO_2_, CL-20/N_2_O, CL-20/NH_2_OH at 2500 K, 3000 K and 3500 K are shown at Figure 11, Figure 12 and Figure 13.

The population of CO_2_ for CL-20/NH_2_OH and CL-20/H_2_O are nearly equivalent at high temperatures. The population of CO_2_ for CL-20/N_2_O is the lowest, while the population of CO_2_ for CL-20/H_2_O is largest at 2500 K. As the temperature increase, the more population of CO_2_ for CL-20/N_2_O grows acutely. However, the population of CO_2_ for CL-20/CO_2_, CL-20/NH_2_OH decrease acutely. It demonstrates that the CO_2_-produced mechanism for a N_2_O guest is different with CO_2_ and NH_2_OH guests. The populations of N_2_ for CL-20/NH_2_OH and CL-20/H_2_O are nearly equivalent at high temperatures. As the temperature increases, the more the population of CO_2_ for CL-20/CO_2_, CL-20/N_2_O grows acutely. It demonstrates that the N_2_-produced mechanism for NH_2_OH guest is different with CO_2_ and N_2_O guests. The populations of H_2_O for CL-20/NH_2_OH and CL-20/H_2_O are nearly equivalent at high temperatures. The populations of H_2_O for CL-20/CO_2_ and CL-20/N_2_O are nearly equivalent at high temperatures. The varying tendency of population for the three main products shows that the influence of guest NH_2_OH and H_2_O, CO_2_ and N_2_O are much same to each other. This may be caused by the hydrogen for two guest groups.

#### 2.4.2. Effect of Nitrogen-Guest on the k_3_

The final products of thermal decomposition of CL-20/guest are N_2_, CO_2_ and H_2_O. The formation rates *k*_3_ can be obtained by fitting the variation trend of the final products with the exponential function [34]: *C*(*t*) = *C*_∞_{1 − exp[−*k*_3_(*t* − *t*_i_)]}, where *C*_∞_ is the asymptotic number of the product, *k*_3_ is the formation rate constant of the product, and *t*_i_ is the time of appearance of the product.

Comparison of the *k*_3_ values of CO_2_, H_2_O and N_2_ for the (a) CL-20/H_2_O, (b) CL-20/CO_2_, (c) CL-20/N_2_O, (d) CL-20/NH_2_OH at different temperatures is shown in Figure 14. All the *k*_3_ values of CO_2_, H_2_O and N_2_ are increased as the temperature improvement. This may be due to the facilitation on production the three main products. The *k*_3_ value of H_2_O is larger than that of N_2_, while the *k*_3_ value of N_2_ is larger than that of CO_2_ for CL-20/H_2_O, CL-20/CO_2_, CL-20/N_2_O, CL-20/NH_2_OH. This demonstrates that the production of H_2_O is the easiest and the production of CO_2_ is the most difficult.

Comparison of the *k*_3_ values of CO_2_, H_2_O and N_2_ for the CL-20/H_2_O, CL-20/CO_2_, CL-20/N_2_O, CL-20/NH_2_OH at different temperatures is shown in Figure 15. For the CL-20/CO_2_ system, the *k*_3_ value of CO_2_ is slightly higher than that for CL-20/H_2_O, CL-20/N_2_O, CL-20/NH_2_OH systems, while the *k*_3_ values of N_2_ and H_2_O are slight smaller than that for CL-20/H_2_O, CL-20/N_2_O, CL-20/NH_2_OH systems. This indicates that CO_2_ restrains the formation of H_2_O and N_2_ molecules. For the CL-20/N_2_O system, the *k*_3_ value of CO_2_ is slightly smaller than that for CL-20/H_2_O, CL-20/CO_2_, CL-20/NH_2_OH systems. This indicates that N_2_O restrains the formation of CO_2_. For the CL-20/NH_2_OH system, the *k*_3_ value of H_2_O is slightly larger than that for CL-20/H_2_O, CL-20/CO_2_, CL-20/N_2_O systems. This indicates that NH_2_OH accelerates the formation of H_2_O.

## 3. Discussion

We have performed MD/ReaxFF-lg simulations to investigate the thermal decomposition reaction of the CL-20/H_2_O, CL-20/CO_2_, CL-20/N_2_O, CL-20/NH_2_OH systems at different temperatures. In this work, guest is not only enhanced the safety but also improved its detonation performance.

During the thermal decomposition reaction of CL-20/H_2_O, CL-20/CO_2_, CL-20/N_2_O, CL-20/NH_2_OH systems at different temperatures, the initial reaction path is not significantly influenced by the incorporation of CO_2_, N_2_O, NH_2_OH: C_6_H_6_O_12_N_12_ → C_6_H_6_O_10_N_11_ + NO_2_ and C_6_H_6_O_12_N_12_ → C_6_H_5_O_12_N_12_ + H. At the same high temperature, the frequencies of both two main initial decomposition reaction are not significantly different for CL-20/H_2_O, CL-20/CO_2_, CL-20/N_2_O, CL-20/NH_2_OH. As for the increase with the temperatures, the frequency of both main initial decomposition reactions improve for CL-20/H_2_O, CL-20/CO_2_, CL-20/N_2_O, CL-20/NH_2_OH. The nitrogen guest can inhibit the trigger decomposition for the larger *E*_a1_ and *E*_a2_ values. Embedding N_2_O and NH_2_OH can significantly accelerate the reaction in the initial decomposition rates at high temperatures for the larger *k*_1_ at high temperatures. However, incorporation of CO_2_, higher temperature has a significant influence on the initial decomposition for the complex *k*_1_. Embedding CO_2_ and N_2_O significantly inhibits the reaction in the intermediate decomposition stage at high temperatures for the smaller *k*_2_ at high temperatures. Incorporation of NH_2_OH has little effect on the reaction rate at high temperatures, with a small difference of *k*_2_. All the *k*_3_ values of CO_2_, H_2_O and N_2_ are increased as the temperature improves. Guest CO_2_ restrains the formation of H_2_O and N_2_ molecules for the higher *k*_3_ value. Guest N_2_O restrains the formation of CO_2_ for the higher *k*_3_ value. Guest NH_2_OH accelerates the formation of H_2_O molecules for the higher *k*_3_ value. The influence of guest NH_2_OH and H_2_O, N_2_O and CO_2_ on decomposition products may be similar for the same amount products. 

The results of this study revealed that the guest CO_2_, N_2_O and NH_2_OH played a certain inhibitory role during the early stages of the host CL-20 thermal decomposition reaction. The study provided a theoretical basis for the synthesis of new energetic materials with host-guest inclusion strategy.

## 4. Computational Methods

The initial unit cell structures of CL-20/H_2_O, CL-20/CO_2_, CL-20/N_2_O and CL-20/NH_2_OH were obtained from the Cambridge Crystallographic Data Centre. In the unit cell, there are eight CL-20 molecules and four guest molecules (H_2_O, CO_2_, N_2_O and NH_2_OH) (Figure 16). We enlarged the unit cell 48 times along both the *a*, the *b* and *c* axes to construct a 6×4×2 supercell containing 48 unit cells ((a) contain 384 of CL-20 and 384 of guest. The supercell of (b), (c), (d) contains 384 of CL-20 and 192 of guest). 

First, the canonical ensemble (NVT) and the Berendsen thermostat were applied to the molecular dynamics (MD) simulation with a total time of 10 ps at 1 K, which further relaxed the α-CL-20/guest supercell. Then, ReaxFF-lg isobaric-isothermal MD (NPT-MD) simulations were performed for 5 ps at 300 K controlled by the Berendsen thermostat based on the relax supercell. Finally, the target temperatures (2500, 2750, 3000, 3250, and 3500 K) are all direct heat from 300 K with NVT-MD simulations. The five different high temperatures are selected to accurately calculate the reaction rate constant and energy barrier. The damping constant is set to 0.25 fs. Komeiji demonstrated that 0.25 fs is enough for calculation accuracy of bonds and angles in molecular dynamics simulations. NVT-MD simulations of the supercell system with the Berendsen thermostat were performed until the potential energy (PE) stabilized. An analysis of the fragments was performed with a 0.3 bond order cutoff value for each atom pair to identify the chemical species [35,36]. The information of the dynamic trajectory was recorded every 20 fs, which was used to analyze the evolution details of α-CL-20/guest in the pyrolysis process.

To verify the suitability of the ReaxFF-lg force field for the CL-20/guest system, we compared the lattice parameters and density of relaxed CL-20/guest at 298 K and 0 Pa with the initial structure from the CCDC (Table 4). The cell parameters and density of relaxed CL-20/guest calculated by MD/ReaxFF-lg agreed well with the initial structure values for the error value < 5%. This preliminarily indicated that ReaxFF-lg can describe the decomposition of CL-20/guest system.

## Figures and Tables

**Figure 1 ijms-24-01840-f001:**
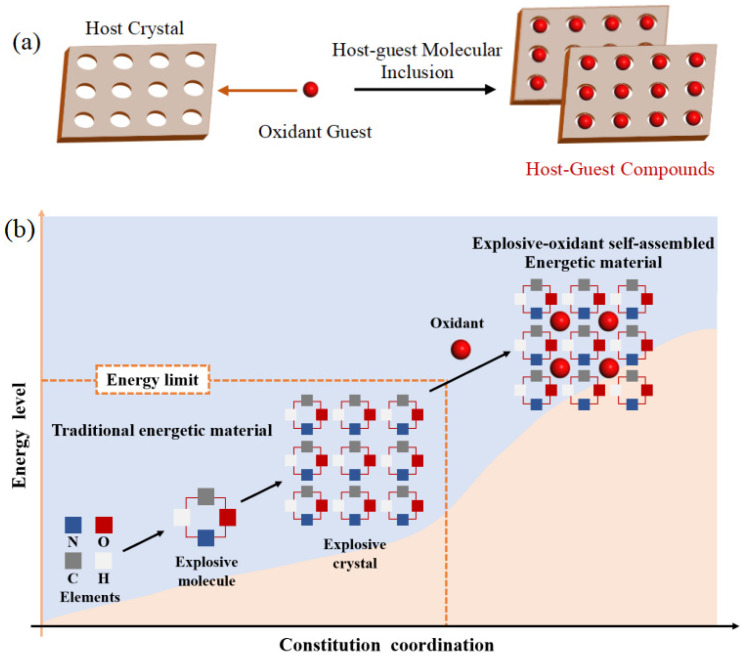
(**a**) 3D energetic host-guest inclusion materials [1]. (**b**) Illustration of explosive-oxidant self-assembled strategy and its comparison with traditional energetic material [2].

**Figure 2 ijms-24-01840-f002:**
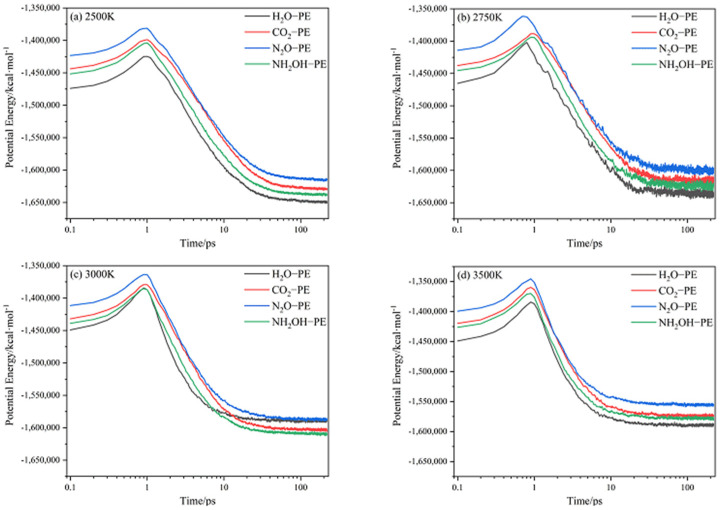
Evolution of potential energy of CL−20/H_2_O, CL−20/CO_2_, CL−20/N_2_O, CL−20/NH_2_OH system with time at (**a**) 2500 K, (**b**) 2750 K, (**c**) 3000 K, (**d**) 3500 K. Thick trendline corresponds to the actual concentration data of corresponding matching color.

**Figure 3 ijms-24-01840-f003:**
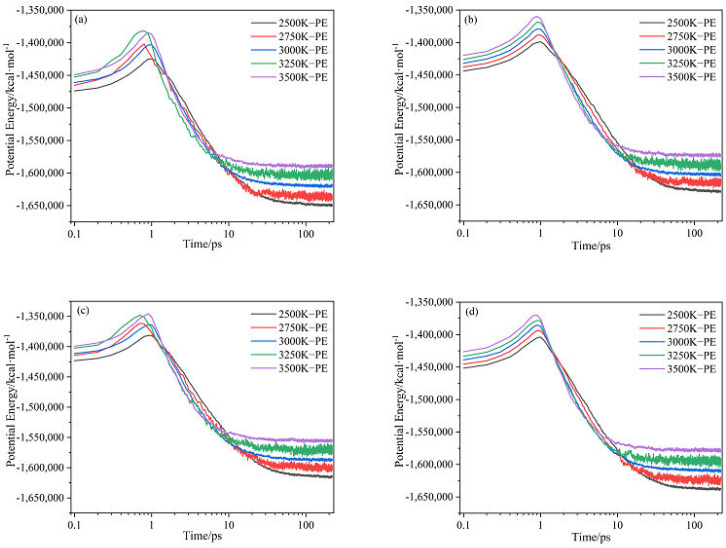
Evolution of potential energy of (**a**) CL-20/H_2_O, (**b**) CL−20/CO_2_, (**c**) CL−20/N_2_O, (**d**) CL−20/NH_2_OH system with time at different temperatures. Thick trendline corresponds to the actual concentration data of corresponding matching color.

**Figure 4 ijms-24-01840-f004:**
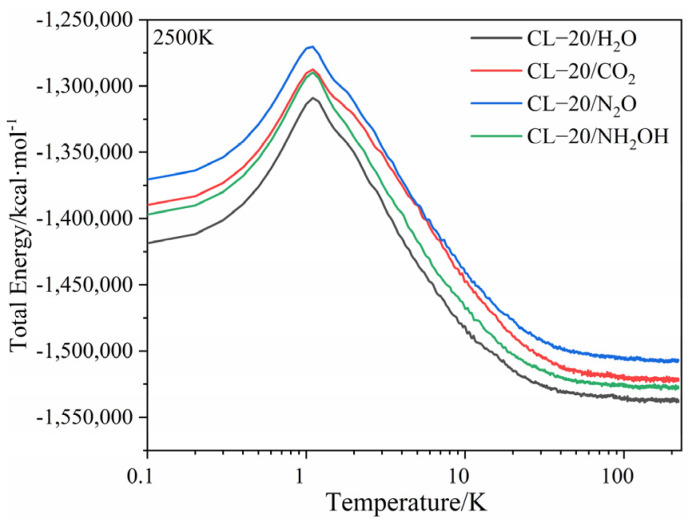
The total energy of CL−20/H_2_O, CL−20/CO_2_, CL−20/N_2_O, CL−20/NH_2_OH systems with time at 2500 K. Thick trendline corresponds to the actual concentration data of corresponding matching color.

**Figure 5 ijms-24-01840-f005:**
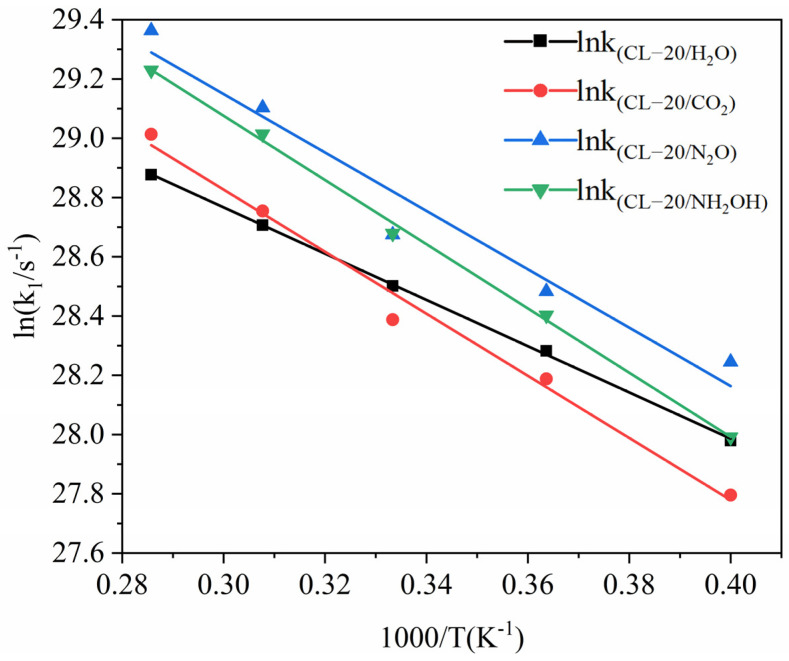
The logarithm of reaction rate (ln(*k*_1_/s^−1^)) against inverse temperature (1/T) in the exothermic decomposition stage at different temperatures. Thick trendline corresponds to the actual concentration data of corresponding matching color.

**Figure 6 ijms-24-01840-f006:**
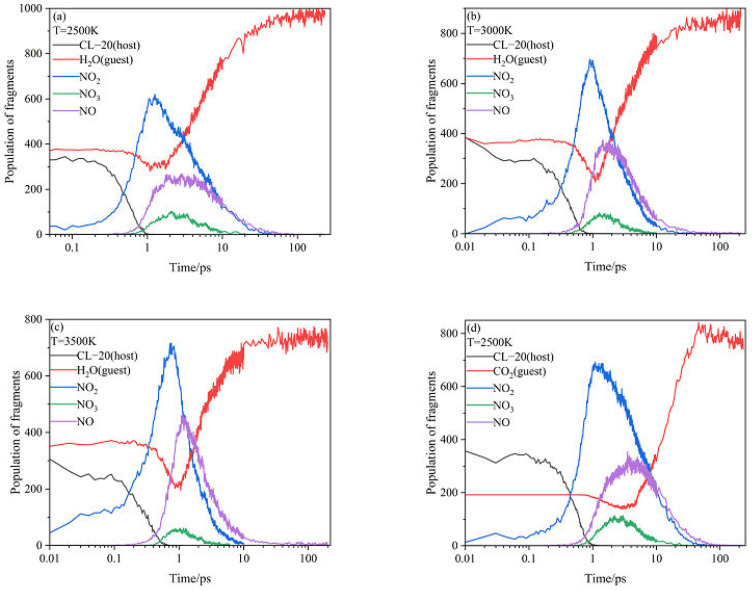
Evolution curves of the main intermediate products and host–guest molecules at different temperatures (**a**–**l**). Thick trendline corresponds to the actual concentration data of corresponding matching color.

**Figure 7 ijms-24-01840-f007:**
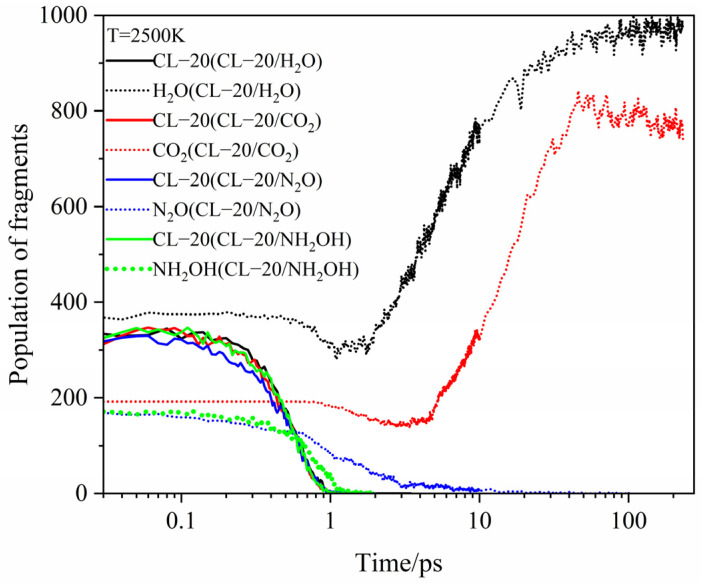
Evolution curves of the host and guest molecules for CL-20/H_2_O, CL-20/CO_2_, CL-20/N_2_O, CL-20/NH_2_OH at 2500 K. Thick trendline corresponds to the actual concentration data of corresponding matching color.

**Figure 8 ijms-24-01840-f008:**
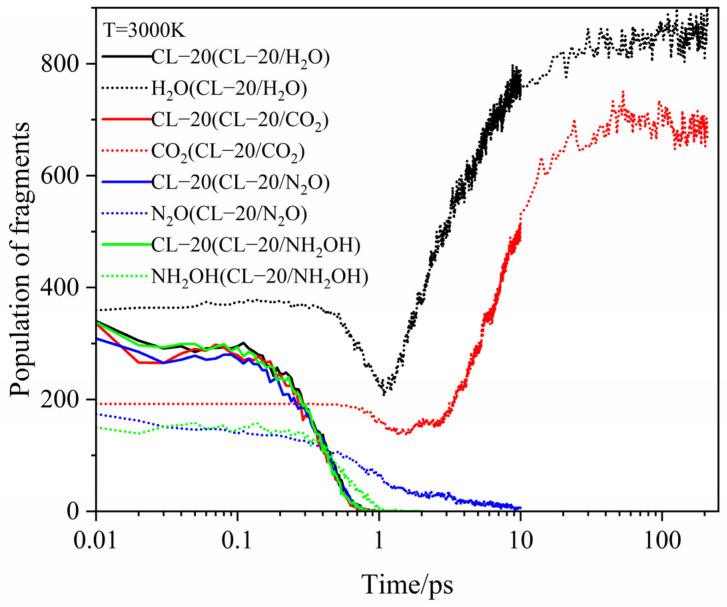
Evolution curves of the host and guest molecules for CL-20/H_2_O, CL-20/CO_2_, CL-20/N_2_O, CL-20/NH_2_OH at 3000 K. Thick trendline corresponds to the actual concentration data of corresponding matching color.

**Figure 9 ijms-24-01840-f009:**
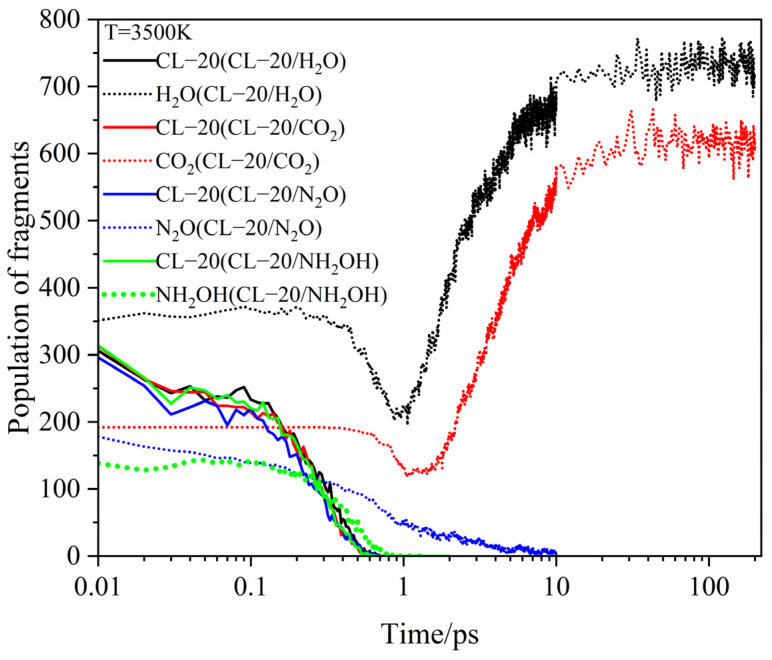
Evolution curves of the host and guest molecules for CL-20/H_2_O, CL-20/CO_2_, CL-20/N_2_O, CL-20/NH_2_OH at 3500 K. Thick trendline corresponds to the actual concentration data of corresponding matching color.

**Figure 10 ijms-24-01840-f010:**
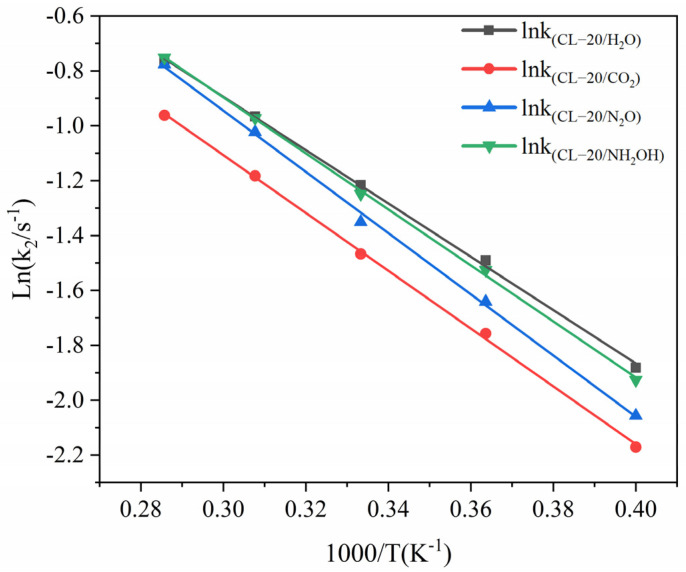
The logarithm of reaction rate (ln(*k*_1_/s^−1^)) against inverse temperature (1/T) in the exothermic decomposition stage at different temperatures. Thick trendline corresponds to the actual concentration data of corresponding matching color.

**Figure 11 ijms-24-01840-f011:**
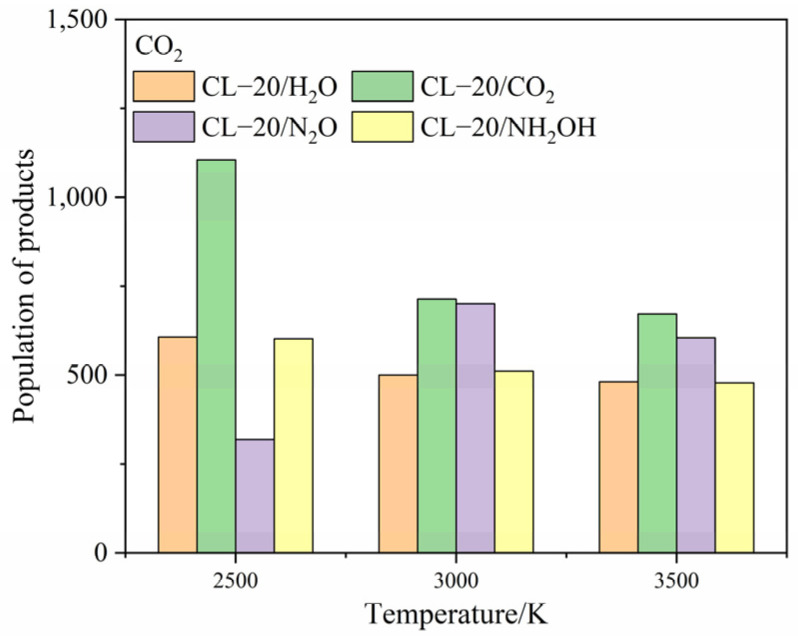
The population of CO_2_ after the complete decomposition reaction for CL-20/H_2_O, CL-20/CO_2_, CL-20/N_2_O, CL-20/NH_2_OH at 2500 K, 3000 K and 3500 K.

**Figure 12 ijms-24-01840-f012:**
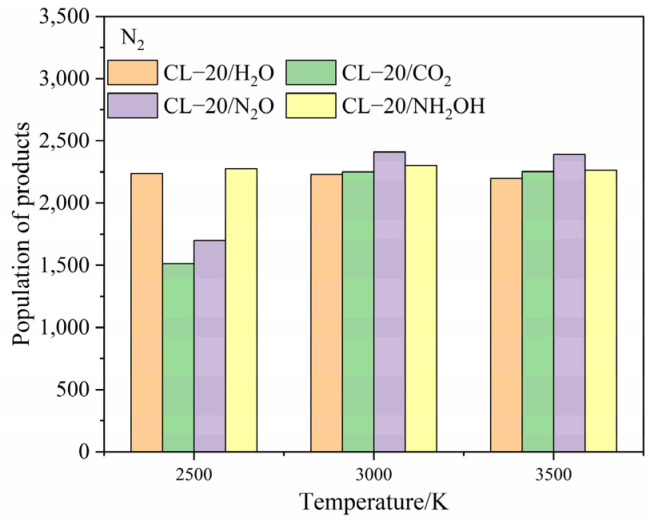
The population of N_2_ after the complete decomposition reaction for CL-20/H_2_O, CL-20/CO_2_, CL-20/N_2_O, CL-20/NH_2_OH at 2500 K, 3000 K and 3500 K.

**Figure 13 ijms-24-01840-f013:**
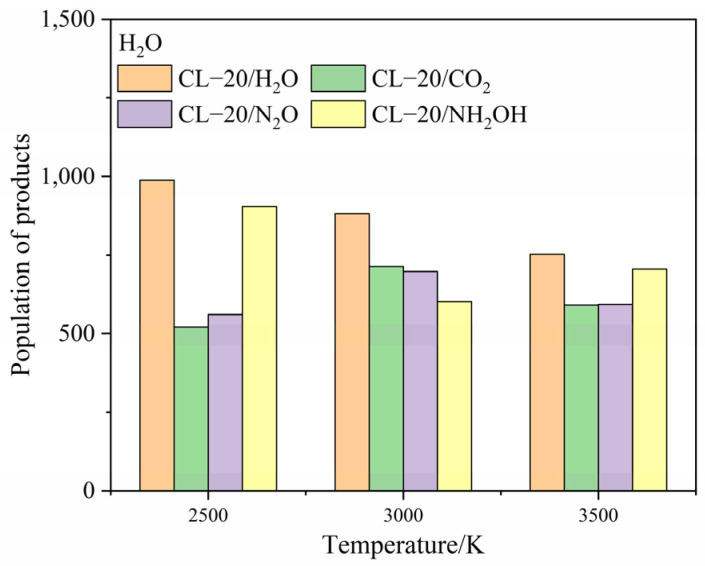
The population of H_2_O after the complete decomposition reaction for CL-20/H_2_O, CL-20/CO_2_, CL-20/N_2_O, CL-20/NH_2_OH at 2500 K, 3000 K and 3500 K.

**Figure 14 ijms-24-01840-f014:**
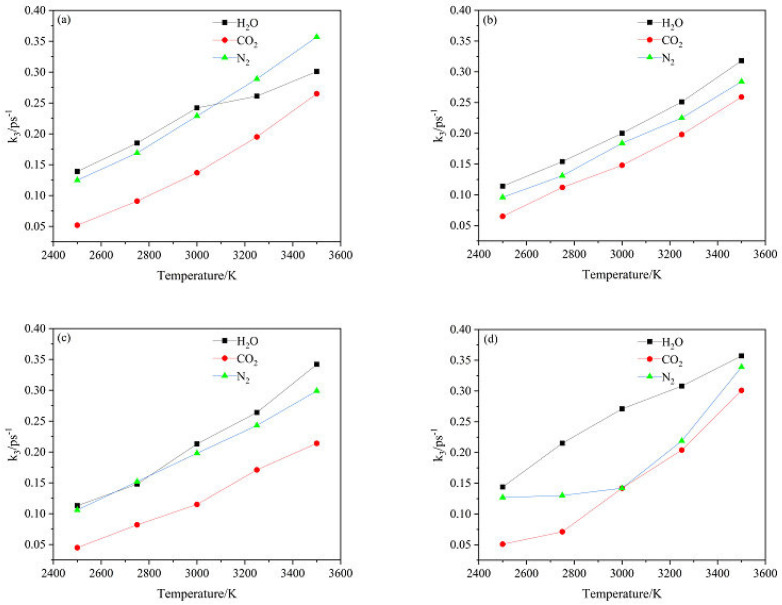
Comparison of the *k*_3_ values of CO_2_, H_2_O and N_2_ for (**a**) CL-20/H_2_O, (**b**) CL-20/CO_2_, (**c**) CL-20/N_2_O, (**d**) CL-20/NH_2_OH at different temperatures. Thick trendline corresponds to the actual concentration data of corresponding matching color.

**Figure 15 ijms-24-01840-f015:**
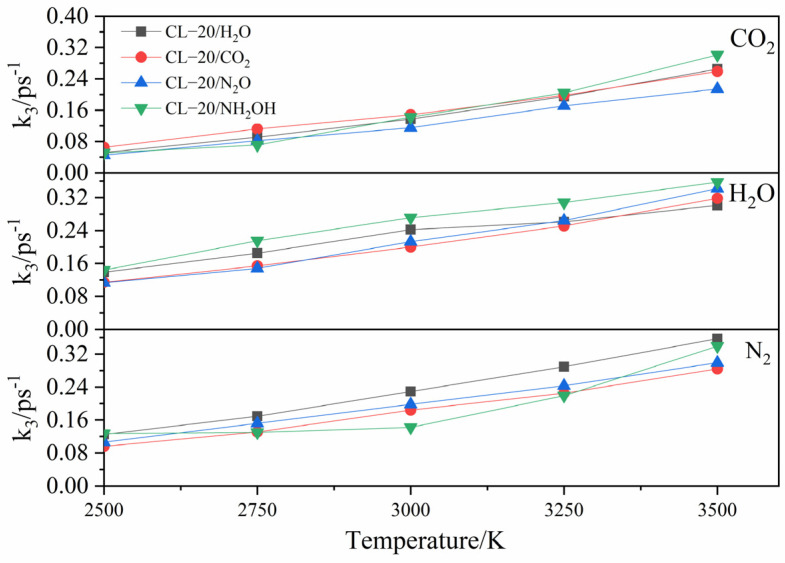
Comparison of the *k*_3_ values of CO_2_, H_2_O and N_2_ for CL-20/H_2_O, CL-20/CO_2_, CL-20/N_2_O, CL-20/NH_2_OH at different temperatures. Thick trendline corresponds to the actual concentration data of corresponding matching color.

**Figure 16 ijms-24-01840-f016:**
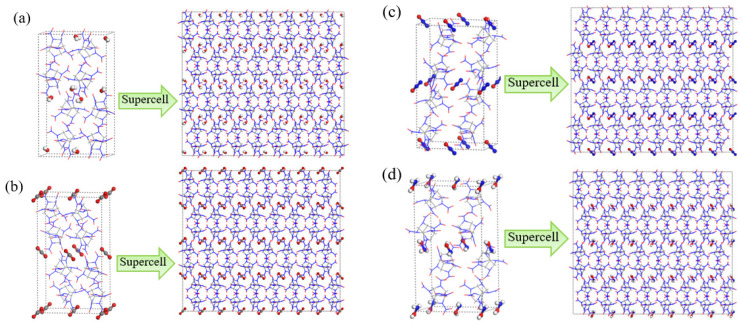
(**a**) α-CL-20/H_2_O, 6 × 4 × 2 α-CL-20/H_2_O supercell, (**b**) α-CL-20/CO_2_, 6 × 4 × 2 α-CL-20/CO_2_ supercell, (**c**) α-CL-20/N_2_O, 6 × 4 × 2 α-CL-20/N_2_O supercell, (**d**) α-CL-20/NH_2_OH, 6 × 4 × 2 α-CL-20/NH_2_OH supercell. The blue atoms represent nitrogen, the red atoms represent oxygen, the white atoms represent hydrogen, the gray atoms represent carbon. The supercell of (**a**) contains 384 of CL-20 and 384 of guest. The supercell of (**b**–**d**) contains 384 of CL-20 and 192 of guest.

**Table 1 ijms-24-01840-t001:** The initial reaction paths of host–guest molecules and their occurrence frequency for CL-20/H_2_O, CL-20/CO_2_, CL-20/N_2_O, CL-20/NH_2_OH at four high temperatures.

Host–Guest Crystal	Temperatures	Initial Reaction Paths	Frequency
CL-20/H_2_O	2500	C_6_H_6_O_12_N_12_ → C_6_H_6_O_10_N_11_ + NO_2_	29
		C_6_H_6_O_12_N_12_ → C_6_H_5_O_12_N_12_ + H	21
		H_2_O → H + OH	20
	3000	C_6_H_6_O_12_N_12_ → C_6_H_6_O_10_N_11_ + NO_2_	51
		C_6_H_6_O_12_N_12_ → C_6_H_5_O_12_N_12_ + H	31
		H_2_O → H + OH	20
	3500	C_6_H_6_O_12_N_12_ → C_6_H_6_O_10_N_11_ + NO_2_	69
		C_6_H_6_O_12_N_12_ → C_6_H_5_O_12_N_12_ + H	41
		H_2_O → H + OH	27
CL-20/CO_2_	2500	C_6_H_6_O_12_N_12_ → C_6_H_6_O_10_N_11_ + NO_2_	41
		C_6_H_6_O_12_N_12_ → C_6_H_5_O_12_N_12_ + H	25
	3000	C_6_H_6_O_12_N_12_ → C_6_H_6_O_10_N_11_ + NO_2_	60
		C_6_H_6_O_12_N_12_ → C_6_H_5_O_12_N_12_ + H	38
	3500	C_6_H_6_O_12_N_12_ → C_6_H_6_O_10_N_11_ + NO_2_	65
		C_6_H_6_O_12_N_12_ → C_6_H_5_O_12_N_12_ + H	34
CL-20/N_2_O	2500	C_6_H_6_O_12_N_12_ → C_6_H_6_O_10_N_11_ + NO_2_	30
		C_6_H_6_O_12_N_12_ → C_6_H_5_O_12_N_12_ + H	25
		N_2_O → N + NO	5
		N_2_O → N_2_ + O	18
	3000	C_6_H_6_O_12_N_12_ → C_6_H_6_O_10_N_11_ + NO_2_	63
		C_6_H_6_O_12_N_12_ → C_6_H_5_O_12_N_12_ + H	31
		N_2_O → N + NO	5
		N_2_O → N_2_ + O	34
	3500	C_6_H_6_O_12_N_12_ → C_6_H_6_O_10_N_11_ + NO_2_	77
		C_6_H_6_O_12_N_12_ → C_6_H_5_O_12_N_12_ + H	49
		N_2_O → N + NO	11
		N_2_O → N_2_ + O	24
CL-20/NH_2_OH	2500	C_6_H_6_O_12_N_12_ → C_6_H_6_O_10_N_11_ + NO_2_	31
		C_6_H_6_O_12_N_12_ → C_6_H_5_O_12_N_12_ + H	22
		NH_2_OH → NH_2_ + OH	8
		NH2OH → NH_2_O + H	11
	3000	C_6_H_6_O_12_N_12_ → C_6_H_6_O_10_N_11_ + NO_2_	41
		C_6_H_6_O_12_N_12_ → C_6_H_5_O_12_N_12_ + H	31
		NH_2_OH → NH_2_ + OH	27
		NH_2_OH → NH_2_O + H	15
	3500	C_6_H_6_O_12_N_12_ → C_6_H_6_O_10_N_11_ + NO_2_	69
		C_6_H_6_O_12_N_12_ C_6_H_5_O_12_N_12_ + H	48
		C_6_H_6_O_10_N_11_ → C_6_H_6_O_8_N_10_ + NO_2_	6
		C_6_H_5_O_12_N_12_ → C_6_H_5_O_10_N_11_ + NO_2_	7
		C_6_H_6_O_12_N_12_ → C_6_H_4_O_12_N_12_ + 2H	5
		NH_2_OH → NH_2_ + OH	35
		NH_2_OH → NH_2_O + H	20

**Table 2 ijms-24-01840-t002:** Reaction rate constant k_1_ in the initial endothermic reaction stage.

Host–Guest Crystal	T/K	k_1_/ps^−1^
CL-20/H_2_O	2500	1.417
	2750	1.918
	3000	2.388
	3250	2.932
	3500	3.476
CL-20/CO_2_	2500	1.179
	2750	1.745
	3000	2.131
	3250	3.075
	3500	3.984
CL-20/N_2_O	2500	1.848
	2750	2.344
	3000	2.839
	3250	4.357
	3500	5.653
CL-20/NH_2_OH	2500	1.434
	2750	2.163
	3000	2.851
	3250	3.985
	3500	4.944

**Table 3 ijms-24-01840-t003:** Partial parameters of PE curve attenuation process.

Host–Guest Crystal	T/K	*U* _max_	*U* _∞_	Δ*U_exo_*	*k*_2_/ps^−1^
CL-20/H_2_O	2500	−1,424,794	−1,650,929	226,135	0.1523
	2750	−1,413,953	−1,636,452	222,499	0.2251
	3000	−1,403,193	−1,621,948	218,755	0.2963
	3250	−1,397,567	−1,606,748	209,181	0.38057
	3500	−1,384,266	−1,591,748	207,482	0.46484
CL-20/CO_2_	2500	−1,398,845	−1,630,254	231,409	0.11404
	2750	−1,389,461	−1,617,052	227,591	0.17252
	3000	−1,379,278	−1,603,764	224,486	0.23059
	3250	−1,3694,15	−1,589,817	220,402	0.30645
	3500	−1,359,754	−1,575,870	216,116	0.38217
CL-20/N_2_O	2500	−1,381,272	−1,616,532	235,260	0.12789
	2750	−1,372,536	−1,602,557	230,021	0.19377
	3000	−1,363,624	−1,588,473	224,849	0.25909
	3250	−1,354,660	−1,572,699	218,039	0.35959
	3500	−1,345,497	−1,556,825	211,328	0.46010
CL-20/NH_2_OH	2500	−1,403,762	−1,639,940	236,178	0.14567
	2750	−1,395,173	−1,625,769	230,596	0.21743
	3000	−1,385,384	−1,611,798	226,414	0.28695
	3250	−1,377,669	−1,595,917	218,248	0.378106
	3500	−1,369,830	−1,580,037	210,207	0.47146

**Table 4 ijms-24-01840-t004:** Comparison of lattice parameters and density of CL-20/guest.

Crystal	Method	*a*/Å	*b*/Å	*c*/Å	*ρ*/g·cm^−3^
CL-20/H_2_O	from CCDC	9.477	13.139	23.380	2.081
	ReaxFF-lg	9.370	12.993	23.119	2.153
CL-20/CO_2_	from CCDC	9.673	13.203	23.553	2.033
	ReaxFF-lg	9.467	13.167	23.489	2.049
CL-20/N_2_O	from CCDC	9.577	13.256	23.625	2.038
	ReaxFF-lg	9.427	13.049	23.256	2.137
CL-20/NH_2_OH	from CCDC	9.789	13.123	23.509	2.000
	ReaxFF-lg	9.602	12.873	23.059	2.119

## Data Availability

Not applicable.

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
