# Peer review of "Effects of Different Guests on Pyrolysis Mechanism of α-CL−20/Guest at High Temperatures by Reactive Molecular Dynamics Simulations at High Temperatures"

_ijms, 2023, doi:10.3390/ijms24031840_

Round 1
Reviewer 1 Report
In this manuscript the authors reported the investigation of analysis the role of guest influence on the pyrolysis decomposition of the host-guest explosive. incorporation of CO2, N2O, NH2OH can inhibit the initial decomposition and intermediate decomposition stage of CL-20/H2O. Some interesting results are obtained. I therefore recommend an acceptance for publishing after next revisions.
1.Pages 2, abstract part, some background sentences can be added;
2.Introduction part, if possible, some important and relative reports about guest influence nanostructures from various styles should be added to show clear background;
3. what about the stability for this Nanocomposite, pleas add more describe?
4. Some minor Language error and style should be modified;
Author Response
Comments and Suggestions for Authors
In this manuscript the authors reported the investigation of analysis the role of guest influence on the pyrolysis decomposition of the host-guest explosive incorporation of CO2, N2O, NH2OH can inhibit the initial decomposition and intermediate decomposition stage of CL-20/H2O. Some interesting results are obtained. I therefore recommend an acceptance for publishing after next revisions.
- Pages 2, abstract part, some background sentences can be added;
Answer: The abstract part, the advantages of CL-20 are highlighted; the polymorphs of CL-20 are introduced; the solvates of CL-20 are also defined after host-guest explosives. The sentence “2,4,6,8,10,12-hexanitro-2,4,6,8,10,12-hexaazaisowurtzitane (CL-20) as one of the energetic materials with the highest density has been widely studied recently.” is changed to be “2,4,6,8,10,12-hexanitro-2,4,6,8,10,12-hexaazaisowurtzitane (CL-20) as one of the energetic materials with greatest detonation performances and highest density from synthesized compounds has been widely studied recently [7].” Then some background sentences are added “CL-20 has five polymorphs (namely β, γ, ε, ζ and δ) and an α-form hydrate [8], in which the ε-CL-20 has the highest density (2.044g·cm-3) [8], β-CL-20 and γ-CL-20 are the dominant configurations of CL-20 molecules found in co-crystals [9]. Thus, many approaches have been used to tune its performance, such as co-crystals [10-13], nanosized particles [14] and so on. The nanoscaling cocrystals demonstrate enhanced stability and high solubility of nano-particles [15]. However, the high production costs and sensitivity (high mechanical sensitivity and easily exhibit phase transition) limit the widespread use of CL-20 [16,17]. Fortunately, removal of H2O from hydrated α-CL-20 can retained the stacking model [18,19]. Then, small molecules are filled in the cavity structure region that removes H2O molecules to form α-CL-20 host-guest explosives [3-5], which also can be regarded as the solvates of CL-20 [20-22].”
- Muravyev, N. V., Wozniak, D. R., Piercey, D. G. Progress and performance of energetic materials: open dataset, tool, and implications for synthesis. J. Mater. Chem. A. 2022, 10, 11054–11073. https://doi.org/10.1039/D2TA01339H.
- Nielsen, A. T., Chafin, A. P., Christian, S. L., Moore, D. W., Nadler, M. P., Nissan, R. A., Vanderah D. J., Gilardi R. D., George C. F., Flippen-Anderson, J. L. Synthesis of polyazapolycyclic caged polynitramines. Tetrahedron1998, 54(39), 11793-11812. https://doi.org/10.1016/S0040-4020(98)83040-8.
- Liu, G., Li, H., Gou, R.J., & Zhang, C. Packing structures of CL-20-based cocrystals. Growth Des.2018, 18(11), 7065-7078. https://doi.org/10.1021/acs.cgd.8b01228.
- Saint Martin, S., Marre, S., Guionneau, P., Cansell, F., Renouard, J., Marchetto, V., Aymonier, C. Host-Guest Inclusion Compound from Nitramine Crystals Exposed to Condensed Carbon Dioxide. Chem.-Eur. J. 2010, 16(45), 13473-13478. https://doi.org/10.1002/chem.201001600.
- Millar, D. I., Maynard-Casely, H. E., Allan, D. R., Cumming, A. S., Lennie, A. R., Mackay, A. J., Oswald, I. D. H., Tang, C. C., Pulham, C. R. Crystal engineering of energetic materials: Co-crystals of CL-20. CrystEngComm 2012, 14(10), 3742-3749. https://doi.org/10.1039/C2CE05796D.
- Bennion, J. C., Siddiqi, Z. R., & Matzger, A. J. A melt castable energetic cocrystal. Chem. Commun. 2017, 53(45), 6065-6068. https://doi.org/10.1039/C7CC02636F.
- Bolton, O., Simke, L. R., Pagoria, P. F., Matzger, A. J. High power explosive with good sensitivity: A 2: 1 cocrystal of CL-20: HMX. Cryst. Growth Des. 2012, 12(9), 4311-4314. https://doi.org/10.1021/cg3010882.
- Han, Q., Zhu, W. Effect of particle size on the thermal decomposition of nano ε-CL-20 by ReaxFF-lg molecular dynamics simulations. Chem. Phys. Lett. 2020, 761, 138067. https://doi.org/10.1016/j.cplett.2020.138067.
- Spitzer, D., Risse, B., Schnell, F., Pichot, V., Klaumünzer, M., Schaefer, M. R. Continuous engineering of nano-cocrystals for medical and energetic applications. Sci. Rep. 2014, 4(1), 1-6. https://doi.org/10.1038/srep06575.
- Simpson, R. L., Urtiew, P. A., Ornellas, D. L., Moody, G. L., Scribner, K. J., Hoffman, D. M. CL‐20 Performance Exceeds that of HMX and its Sensitivity is Moderate. Explos. Pyrot. 1997, 22(5), 249-255. https://doi.org/10.1002/prep.19970220502.
- van der Heijden, A. E., Bouma, R. H. Crystallization and Characterization of RDX, HMX, and CL-20. Growth Des. 2004, 4(5), 999-1007. https://doi.org/10.1021/cg049965a.
- Pu, L., Xu, J., Song, G., Tian, Y., Zhang, H., Liu, X., & Sun, J. Investigation on the thermal expansion of α-CL-20 with different water contents. J. Therm. Anal. Calorim. 2015, 122(3), 1355-1364. https://doi.org/10.1007/s10973-015-4884-6.
- Wang, Y., Song, S., Huang, C., Qi, X., Wang, K., Liu, Y., Zhang, Q. Hunting for advanced high-energy-density materials with well-balanced energy and safety through an energetic host-guest inclusion strategy. J. Mater. Chem. A. 2019, 7(33), 19248-19257. https://doi.org/10.1039/C9TA04677A.
- Fedyanin, I. V., Lyssenko, K. A., Fershtat, L. L., Muravyev, N. V., Makhova, N. N. Crystal Solvates of Energetic 2, 4, 6, 8, 10, 12-Hexanitro-2, 4, 6, 8, 10, 12-hexaazaisowurtzitane Molecule with [bmim]-Based Ionic Liquids. Cryst. Growth Des. 2019, 19(7), 3660-3669. https://doi.org/10.1021/acs.cgd.8b01835.
- Zharkov, M. N., Kuchurov, I. V., Zlotin, S. G. Micronization of CL-20 using supercritical and liquefied gases. CrystEngComm 2020, 22(44), 7549-7555. https://doi.org/10.1039/D0CE01167C.
- Yudin, N. V., Sinditskii, V. P., Filatov, S. A., Serushkin, V. V., Kostin, N. A., Ivanyan, M. V., Zhang, J. G. Solvate of 2, 4, 6, 8, 10, 12-Hexanitro-2, 4, 6, 8, 10, 12-Hexaazaisowurtzitane (CL-20) with both N2O4 and Stable NO2 Free Radical. ChemPlusChem 2020, 85(9), 1994-2000. https://doi.org/10.1002/cplu.202000534.
- Introduction part, if possible, some important and relative reports about guest influence nanostructuresfrom various styles should be added to show clear background;
Answer: The sentence “The nanoscaling cocrystals demonstrate enhanced stability and high solubility of nano-particles [15].” is added.
- Spitzer, D., Risse, B., Schnell, F., Pichot, V., Klaumünzer, M., Schaefer, M. R. Continuous engineering of nano-cocrystals for medical and energetic applications. Rep. 2014, 4(1), 1-6. https://doi.org/10.1038/srep06575.
- what about the stability for this Nanocomposite, pleas add more describe?
Answer: The sentence “Thus, many approaches have been used to tune its performance, such as co-crystals [10-13], nanosized particles [14] and so on.” is added.
- Saint Martin, S., Marre, S., Guionneau, P., Cansell, F., Renouard, J., Marchetto, V., Aymonier, C. Host-Guest Inclusion Compound from Nitramine Crystals Exposed to Condensed Carbon Dioxide. -Eur. J. 2010, 16(45), 13473-13478. https://doi.org/10.1002/chem.201001600.
- Millar, D. I., Maynard-Casely, H. E., Allan, D. R., Cumming, A. S., Lennie, A. R., Mackay, A. J., Oswald, I. D. H., Tang, C. C., Pulham, C. R. Crystal engineering of energetic materials: Co-crystals of CL-20. CrystEngComm 2012, 14(10), 3742-3749. https://doi.org/10.1039/C2CE05796D.
- Bennion, J. C., Siddiqi, Z. R., & Matzger, A. J. A melt castable energetic cocrystal. Chem. Commun. 2017, 53(45), 6065-6068. https://doi.org/10.1039/C7CC02636F.
- Bolton, O., Simke, L. R., Pagoria, P. F., Matzger, A. J. High power explosive with good sensitivity: A 2: 1 cocrystal of CL-20: HMX. Cryst. Growth Des. 2012, 12(9), 4311-4314. https://doi.org/10.1021/cg3010882.
- Han, Q., Zhu, W. Effect of particle size on the thermal decomposition of nano ε-CL-20 by ReaxFF-lg molecular dynamics simulations. Phys. Lett.2020, 761, 138067. https://doi.org/10.1016/j.cplett.2020.138067.
- Some minor Language error and style should be modified;
Answer: The sentence “The guest molecules can not only enhance the detonation performance of host explosives but can also enhance their stability.” is changed to be “The guest molecules can not only enhance the detonation performance of host explosives but also can enhance their stability.”
The sentence ”Guest molecules get heavily involved in the reaction and influence on the reaction rate.” is changed to be “Guest molecules get heavily involved in the reaction and influence on the reaction rates.”
The sentence “Thereafter, research focused on adopting a host-guest inclusion strategy to embed suitable guest within the void by removing water of α-CL-20/H2O.” is changed to be “Thereafter, researches are focused on adopting a host-guest inclusion strategy to embed suitable guest within the void by removing water of αα-CL-20/H2O.”

Reviewer 2 Report
Presented manuscript entitled "Effects of different guests on pyrolysis mechanism of α-CL-20/guest at high temperatures by reactive molecular dynamics 3 simulations at high temperatures" discussed theoretically the mechanism of decomposition of CL-20-based energetic composites. THe topic is actual and will be of interest for community, i have only minor comments to improve the problem statement and discussion.
First, to streghten the problem statement, it should be mentioned that CL-20 has one of the greatest detonation performances from synthesized compounds (#5 according to most recent compendium https://doi.org/10.1039/D2TA01339H)
Second, the alpha-phase is not the most commonly studied (in practice, epsilon-phase is more interesting), please discuss it in the introduction
Third, there are several publications on the topic - previously reported solvates of CL-20, please discuss it in the introduction https://doi.org/10.1002/cplu.202000534, https://doi.org/10.1039/D0CE01167C, https://doi.org/10.1021/acs.cgd.8b01835
Author Response
Comments and Suggestions for Authors
Presented manuscript entitled "Effects of different guests on pyrolysis mechanism of α-CL-20/guest at high temperatures by reactive molecular dynamics 3 simulations at high temperatures" discussed theoretically the mechanism of decomposition of CL-20-based energetic composites. The topic is actual and will be of interest for community, I have only minor comments to improve the problem statement and discussion.
First, to streghten the problem statement, it should be mentioned that CL-20 has one of the greatest detonation performances from synthesized compounds (#5 according to most recent compendium https://doi.org/10.1039/D2TA01339H)
Answer: The sentence “2,4,6,8,10,12-hexanitro-2,4,6,8,10,12-hexaazaisowurtzitane (CL-20) as one of the energetic materials with the highest density has been widely studied recently.” is changed to be “2,4,6,8,10,12-hexanitro-2,4,6,8,10,12-hexaazaisowurtzitane (CL-20) as one of the energetic materials with greatest detonation performances and highest density from synthesized compounds has been widely studied recently [7].”
Due to the addition of reference 7, changes in the serial numbers of other relevant references are marked in blue.
- Muravyev, N. V., Wozniak, D. R., Piercey, D. G. Progress and performance of energetic materials: open dataset, tool, and implications for synthesis. J. Mater. Chem. A. 2022, 10, 11054–11073. https://doi.org/10.1039/D2TA01339H.
Second, the alpha-phase is not the most commonly studied (in practice, epsilon-phase is more interesting), please discuss it in the introduction
Answer: The sentence “ CL-20 has five polymorphs (namely β, γ, ε, ζ and δ) and an α-form hydrate [8], in which the ε-CL-20 has the highest density (2.044g·cm-3) [8], β-CL-20 and γ-CL-20 are the dominant configurations of CL-20 molecules found in co-crystals [9].” is added after the sentence “2,4,6,8,10,12-hexanitro-2,4,6,8,10,12-hexaazaisowurtzitane (CL-20) as one of the energetic materials with greatest detonation performances and highest density from synthesized compounds has been widely studied recently [7].”
Due to the addition of references, changes in the serial numbers of other relevant references are marked in blue.
- Nielsen, A. T., Chafin, A. P., Christian, S. L., Moore, D. W., Nadler, M. P., Nissan, R. A., Vanderah D. J., Gilardi R. D., George C. F., Flippen-Anderson, J. L. Synthesis of polyazapolycyclic caged polynitramines. Tetrahedron1998, 54(39), 11793-11812. https://doi.org/10.1016/S0040-4020(98)83040-8.
- Liu, G., Li, H., Gou, R.J., & Zhang, C. Packing structures of CL-20-based cocrystals. Growth Des.2018, 18(11), 7065-7078. https://doi.org/10.1021/acs.cgd.8b01228.
Third, there are several publications on the topic - previously reported solvates of CL-20, please discuss it in the introduction https://doi.org/10.1002/cplu.202000534, https://doi.org/10.1039/D0CE01167C, https://doi.org/10.1021/acs.cgd.8b01835
Answer: CL-20 host-guest explosives also can be regarded as the solvates of CL-20, so the solvates of CL-20 would be introduced after CL-20 host-guest explosives. The sentence “which also can be regarded as the solvates of CL-20 [20-22]” is added after the sentence “Then, small molecules are filled in the cavity structure region that removes H2O molecules to form α-CL-20 host-guest explosives [3-5]”
Due to the addition of references, changes in the serial numbers of other relevant references are marked in blue.
- Fedyanin, I. V., Lyssenko, K. A., Fershtat, L. L., Muravyev, N. V., Makhova, N. N. Crystal Solvates of Energetic 2, 4, 6, 8, 10, 12-Hexanitro-2, 4, 6, 8, 10, 12-hexaazaisowurtzitane Molecule with [bmim]-Based Ionic Liquids. Cryst. Growth Des. 2019, 19(7), 3660-3669. https://doi.org/10.1021/acs.cgd.8b01835.
- Zharkov, M. N., Kuchurov, I. V., Zlotin, S. G. Micronization of CL-20 using supercritical and liquefied gases. CrystEngComm 2020, 22(44), 7549-7555. https://doi.org/10.1039/D0CE01167C.
- Yudin, N. V., Sinditskii, V. P., Filatov, S. A., Serushkin, V. V., Kostin, N. A., Ivanyan, M. V., Zhang, J. G. Solvate of 2, 4, 6, 8, 10, 12-Hexanitro-2, 4, 6, 8, 10, 12-Hexaazaisowurtzitane (CL-20) with both N2O4 and Stable NO2 Free Radical. ChemPlusChem 2020, 85(9), 1994-2000. https://doi.org/10.1002/cplu.202000534.
